# COVID-19 and *C. auris*: A Case-Control Study from a Tertiary Care Center in Lebanon

**DOI:** 10.3390/microorganisms10051011

**Published:** 2022-05-11

**Authors:** Fatima Allaw, Sara F. Haddad, Nabih Habib, Pamela Moukarzel, Nour Sabiha Naji, Zeina A. Kanafani, Ahmad Ibrahim, Nada Kara Zahreddine, Nikolaos Spernovasilis, Garyphallia Poulakou, Souha S. Kanj

**Affiliations:** 1Division of Infectious Diseases, Internal Medicine Department, American University of Beirut Medical Center, Beirut 1107 2020, Lebanon or fballaw@gmail.com (F.A.); sarahaddad711@gmail.com (S.F.H.); zk10@aub.edu.lb (Z.A.K.); 2School of Medicine, American University of Beirut, Beirut 1107 2020, Lebanon; nfh08@mail.aub.edu (N.H.); psm03@mail.aub.edu (P.M.); nsn09@mail.aub.edu (N.S.N.); 3Infection Control and Prevention Program, American University of Beirut Medical Center, Beirut 1107 2020, Lebanon; ai07@aub.edu.lb (A.I.); nk13@aub.edu.lb (N.K.Z.); 4School of Medicine, University of Crete, 71003 Heraklion, Greece; nikspe@hotmail.com; 5German Oncology Center, Limassol 4108, Cyprus; 6ESCMID (European Society of Clinical Microbiology and Infectious Diseases) Study Group for Infections in Critically Ill Patients—ESGCIP, 4001 Basel, Switzerland; 73rd Department of Medicine, Medical School, National and Kapodistrian University of Athens, Sotiria General Hospital, 11572 Athens, Greece

**Keywords:** *Candida auris*, COVID-19, pandemic, central venous catheters, urinary catheter, tocilizumab, length of stay, qSOFA, candida score, infection control

## Abstract

Many healthcare centers around the world have reported the surge of *Candida auris* (*C. auris*) outbreaks during the COVID-19 pandemic, especially among intensive care unit (ICU) patients. This is a retrospective study conducted at the American University of Beirut Medical Center (AUBMC) between 1 October 2020 and 15 June 2021, to identify risk factors for acquiring *C. auris* in patients with severe COVID-19 infection and to evaluate the impact of *C. auris* on mortality in patients admitted to the ICU during that period. Twenty-four non-COVID-19 (COV−) patients were admitted to ICUs at AUBMC during that period and acquired *C. auris* (*C. auris*+/COV−). Thirty-two patients admitted with severe COVID-19 (COV+) acquired *C. auris* (*C. auris*+/COV+), and 130 patients had severe COVID-19 without *C. auris* (*C. auris−*/COV+). Bivariable analysis between the groups of (*C. auris*+/COV+) and (*C. auris−*/COV+) showed that higher quick sequential organ failure assessment (qSOFA) score (*p* < 0.001), prolonged length of stay (LOS) (*p* = 0.02), and the presence of a urinary catheter (*p* = 0.015) or of a central venous catheter (CVC) (*p* = 0.01) were associated with positive culture for *C. auris* in patients with severe COVID-19. The multivariable analysis showed that prolonged LOS (*p* = 0.008) and a high qSOFA score (*p* < 0.001) were the only risk factors independently associated with positive culture for *C. auris*. Increased LOS (*p* = 0.02), high “Candida score” (*p* = 0.01), and septic shock (*p* < 0.001) were associated with increased mortality within 30 days of positive culture for *C. auris*. Antifungal therapy for at least 7 days (*p* = 0.03) appeared to decrease mortality within 30 days of positive culture for *C. auris*. Only septic shock was associated with increased mortality in patients with *C. auris* (*p* = 0.006) in the multivariable analysis. *C. auris* is an emerging pathogen that constitutes a threat to the healthcare sector.

## 1. Introduction

Patients with severe COVID-19 infection usually require a prolonged hospital stay, putting them at risk for acquiring multidrug-resistant organisms (MDRO) [1]. Various fungal superinfections have also been reported with COVID-19 infected patients [2,3]. Colonization with resistant bacteria, intake of steroids and immunomodulators, and prolonged hospital stay have all been described as risk factors and predictors for superimposed bacterial and fungal infections in this patient population [4,5].

*Candida auris* (*C. auris*) has been reported as an emergent fungus which represents a global threat because of its multidrug resistance, laboratory misidentification, and ability to cause outbreaks in healthcare settings with nosocomial dissemination [5]. It has so far been reported from over 30 countries and has constituted a heavy burden to the healthcare system [5]. Since the beginning of the COVID-19 pandemic, outbreaks of *C. auris* have been described from many centers in critically-ill patients [6,7,8,9,10]. The first *C. auris* case ever described in Lebanon occurred during the COVID-19 pandemic at the American University of Beirut Medical Center (AUBMC), a tertiary referral center, where an outbreak with 14 patients with positive cultures for *C. auris* was initially identified over a period of 13 weeks, starting from October 2020. Half of them had COVID-19 infection [11]. Since then, and until June 2021, we have isolated *C. auris* from various sites in 32 COVID-19 patients as well as from 24 patients without COVID-19 infection in different intensive care units (ICU) despite all the interventions initiated by the Infection Prevention and Control (IPC) team to contain the outbreak.

The available literature emphasizes on the methods of *C. auris* identification, its susceptibilities [12,13,14,15], and incriminated risk factors associated with its isolation [16,17,18]. Several reports have described the association between *C. auris* infection and COVID-19 [19,20]. To better understand this association, we conducted a case-control study at AUBMC between October 2020 and June 2021. We herein describe the clinical characteristics of patients with COVID-19 infected with *C. auris* admitted to the ICU. We also evaluated the risk factors of isolating *C. auris* in patients with severe COVID-19 and the outcomes of the infected patients.

## 2. Materials and Methods

### 2.1. Study Design and Data Collection

This is a retrospective case-control study conducted at AUBMC, a teaching hospital with 364 beds and 50 adult ICU beds in 6 different critical care units. Patients who were included were adults 18-years-old and above who had severe COVID-19, or positive *C. auris* culture, or both, and were admitted to any critical care unit at AUBMC between 1 October 2020, and 15 June 2021. These units included COVID-19 ICU, medical and surgical ICU, coronary care unit (CCU), neurology ICU, respiratory care unit (RCU), emergency department (ED) unit (which served as a critical unit during some phases of the pandemic because of the burden on hospital beds), and a step-down unit for continuation of care in patients requiring close monitoring. 

A case-control study was conducted to identify potential risk factors for isolation of *C. auris* in cultures in patients with severe COVID-19 infection: the control group included patients with severe COVID-19 requiring ICU admission because of an increase in oxygen requirements or because of hemodynamic instability. The case group included patients who had severe COVID-19 infection and in whom *C. auris* was isolated during their hospital stay until their discharge or death. COVID-19 was confirmed either by a reverse transcriptase polymerase-chain-reaction (RT-PCR) cobas-Roche^®^ SARS-CoV-2 Test or by a rapid antigen test, Liat^®^, GeneXpert^®^, and BioFire^®^.

Another case-control analysis was done to evaluate the impact of *C. auris* on the mortality of patients admitted to the ICU during that period, regardless of their COVID-19 status. The case group included *C. auris* patients who died within 30 days from *C. auris* isolation, and the control group included *C. auris* patients who were still alive at day 30. All patients with isolated *C. auris* who were admitted to the critical care units during this time period were analyzed including patients with and without COVID-19 infection. No sampling was done in our study and all patients who fit the inclusion criteria during our study period were included. 

*C. auris* was cultured from urine, deep tracheal aspirates (DTA), blood, and wounds. These sites of isolation were cultured when a patient spiked a fever and underwent an infectious work-up. We excluded cases when *C. auris* was only isolated from the skin screening part of the IPC investigation of the outbreak. Except for *C. auris* isolated from blood, it was difficult to tell whether isolates from urine, skin, and DTA were colonizers or real pathogens contributing to an infection in the patients who had prolonged ICU stay and other complications from COVID-19. Additionally, in the case of multiple positive cultures from the same patient, the date of the first positive culture was considered as the date of isolation. 

Data were collected retrospectively from electronic health records. Variables related to patients’ demographics, clinical characteristics, and COVID-19 treatments were reviewed. They included: age, gender, length of hospital stay (LOS), quick sequential organ failure assessment (qSOFA) score at the time of *C. auris* isolation or qSOFA score at the time of ICU admission for COVID-19 patients without *C. auris*, Charlson score, Candida score [21] at the time of *C. auris* isolation, underlying comorbidities (diabetes mellitus type II (DMII), chronic kidney disease (CKD), chronic lung disease, active hematological malignancy, active solid malignancy, active immunosuppressive treatment other than chemotherapy, chemotherapy, previous surgery from 30 days to 48 h of *C. auris* isolation), recent invasive and non-invasive devices (central venous catheter (CVC), urinary catheter, use of endotracheal intubation (ET), nasogastric tube (NG) and mechanical ventilation (MV), use of non-invasive ventilation (Bilevel positive airway pressure (BIPAP) or high flow nasal cannula (HFNC)), previous intake of tocilizumab, baricitinib, prednisone >10 mg per day (for at least one week), and recent hemodialysis and parenteral nutrition, all from 30 days to 48 h prior to *C. auris* isolation. We also looked at the antibiotics administered and the multi-drug resistant (MDR) bacteria cultured in the last 4 weeks prior to *C. auris* isolation including Vancomycin-resistant enterococci (VRE), Carbapenem-resistant Enterobacterales (CRE), methicillin-resistant *Staphylococcus aureus* (MRSA), MDR *Acinetobacter baumannii*, MDR *Pseudomonas aeruginosa*, and Extended-spectrum-β-lactamase (ESBL) producing Enterobacterales. Data on intake of antifungal therapy were collected. Early in-hospital mortality was defined as death within 30 days from *C. auris* isolation. We also looked at septic shock (definition of septic shock was based on the ICU medical team’s clinical diagnosis) and home discharge within 30 days of *C. auris* isolation. As per the existing MDRO management policy implemented at AUBMC, microbiological eradication was considered when two consecutive cultures, including those from skin screening, taken 48 h apart while the patient is not receiving anti-fungal treatment, were negative. Subsequently, a new IPC policy addressed the management of patients with *C. auris.*

The study protocol was approved by the AUBMC Institutional Review Board (IRB) under number BIO-2021-0180 and informed consents were waived due to the retrospective nature of the study.

### 2.2. Statistical Analysis

Categorical variables are presented as frequencies and percentages. Continuous variables are presented as means and standard deviations (SD). Risk factors for acquiring *C. auris* among patients with severe COVID-19 infection were determined and analyzed using bivariate analysis. The association between *C. auris* isolation and other categorical variables was assessed using the Chi-square and Fisher’s test. The Student’s *t*-test was used for the association with continuous variables. A similar analysis was done to assess the variables associated with mortality among all patients with *C. auris* and admitted to any ICU. Logistic regression was used to investigate potential risk factors for isolating *C. auris*, and factors affecting mortality in patients with *C. auris*. The variables that were previously shown in the literature to increase the risk of *C. auris* isolation and to increase mortality in patients with *C. auris* were included in the multivariable model. A *p*-value < 0.05 was considered significant. The collected data were analyzed through Statistical Package for the Social Sciences (SPSS), version 28. 

### 2.3. C. auris Speciation, Identification, and Molecular Characterization

Clinical isolates were cultured on Sabouraud Dextrose Agar or subcultured on chocolate agar medium prior to identification. *C. auris* colonies were phenotypically identified by Matrix-Assisted Laser Desorption/Ionization-Time of Flight MALDI-TOF (Bruker Daltonik, GmbH, Bremen, Germany) and by the VITEK-2 system (bioMérieux, Marcy l’Étoile, France). Samples were saved and referred for genome sequencing analysis in order to reveal the clade distribution and antifungal resistance genes [22]. This was done by AUBMC microbiology laboratory team in collaboration with the Faculty of Medicine and University Hospital in Plzen Czechia [22]. Antifungal susceptibility testing was done using VITEK-2 and E-test antimicrobial susceptibility tests. The microbiology laboratory used the Centers for Disease Control and Prevention (CDC) tentative MIC cut-off values that are based on data from similar *Candida* species [23] to interpret strains’ susceptibilities against amphotericin B, caspofungin, micafungin, fluconazole, and voriconazole. The resistance breakpoints were defined as follows: fluconazole, ≥32 μg/mL; caspofungin, ≥2 μg/mL; micafungin, ≥4 μg/mL; amphotericin B, ≥2 μg/mL. Resistant breakpoints to voriconazole were extrapolated from fluconazole. Due to the financial issues in view of the economic crisis in Lebanon and the COVID-19 pandemic, susceptibility tests could not be performed on several patients. Genomes were determined by whole genome sequencing (WGS) using long reads sequencing (PacBIO) to identify the antifungal resistance genes and the clade distribution [22].

## 3. Results

### 3.1. Study Population and Demographics

Our population is classified into 3 groups as follows: Group (*C. auris*+/COV−) is comprised of patients without COVID-19 infection who were admitted to any ICU and acquired *C. auris* (n = 24); Group (*C. auris*+/COV+) includes COVID-19 patients who were admitted to the ICU and acquired *C. auris* (n = 32), and Group (*C. auris−*/COV+) included all the patients who had severe COVID-19 requiring ICU admission but without isolation of *C. auris* during their hospitalization (n = 130). These patients were admitted to all critical care units of AUBMC between October 1, 2020, and June 15, 2021. 

The demographics and clinical characteristics of the study populations are summarized in Table 1. The median age of patients in Group (*C. auris*+/COV−) was 70.5 (±14.9) years with 41.7% males, 68.4 (±14.2) years in Group (*C. auris*+/COV+) with 59.4% males, and 69.5 (±12.2) years in Group (*C. auris−*/COV+) with 70% males. The mean time between admission and *C. auris* isolation in ICU was 26.3 (±14.4) days for Group (*C. auris*+/COV−) and 32.8 (±19.7) days for Group (*C. auris*+/COV+). The median length of stay (LOS) was 42.0, 51.5, and 22.5 days for patients with *C. auris*, *C. auris* and COVID-19, and COVID-19 alone respectively. There was a significant difference in the LOS between Groups (*C. auris*+/COV−) and (*C. auris−*/COV+) (*p* = 0.003), and Groups (*C. auris*+/COV+) and (*C. auris−*/COV+) (*p* < 0.001), but not between Groups (*C. auris*+/COV−) and (*C. auris*+/COV+) (*p* = 0.13). The mortality in Groups (*C. auris*+/COV−), (*C. auris*+/COV+), and (*C. auris−*/COV+) were 15 (62.5%), 19 (59.4%), and 94 (72.3%) (*p* = 0.29).

qSOFA score on the day of *C. auris* isolation was 1.58 (±0.78) and 1.63 (±0.79) for Groups (*C. auris*+/COV−) and (*C. auris*+/COV+) respectively, and 1.06 (±0.84) for Group (*C. auris−*/COV+) on the day of admission to ICU, with a statistical significance noted between Groups (*C. auris*+/COV+) and (*C. auris−*/COV+) only (*p* = 0.003). Charlson’s score was 5.38 (±2.16), 5.00 (±2.87), and 4.54 (±2.48) for Group (*C. auris*+/COV−), (*C. auris*+/COV+), and (*C. auris−*/COV+) respectively, with no statistical significance, and Candida score was 2.42 (±1.38), 1.59 (±0.98), and 1.39 (±1.17) for Group (*C. auris*+/COV−), (*C. auris*+/COV+), and (*C. auris−*/COV+) respectively, with a significant difference between Groups (*C. auris*+/COV−) and (*C. auris*+/COV+) (*p* = 0.002), and (*C. auris*+/COV−) and (*C. auris−*/COV+) (*p* = 0.012).

For Groups (*C. auris*+/COV−) and (*C. auris*+/COV+), *C. auris* was cultured from DTA in 24 patients (42.86%), from urine in 20 patients (35.57%), from blood in 6 patients (10.71%), and from skin and wounds in 4 patients (7.14%); 2 patients grew *C. auris* (3.57%) from DTA and blood simultaneously. Concerning patients’ locations when the first positive *C. auris* culture was isolated, 5.4% of the patients were in the ED, 7.1% were in COVID ICU, and 87.5% were in non-COVID ICUs (regular ICU, neuro-ICU, CCU, RCU, and step-down unit).

### 3.2. Risk Factors for C. auris in Severe COVID-19

Potential risk factors for isolating *C. auris* from patients with severe COVID-19 infection were analyzed using *t*-test, chi-square, and Fischer, and are presented in Table 2. 

Bivariable analysis of Groups (*C. auris*+/COV+) and (*C. auris−*/COV+) showed that the following variables were associated with acquiring *C. auris*: higher qSOFA score (1.63 vs. 1.06) (*p* < 0.001), presence of a CVC (93.8% vs. 72.3%) (*p* = 0.01), presence of a urinary catheter (100% vs. 85.4%) (*p* = 0.015), and longer hospital length of stay (35.3 vs. 27.0 days) (*p* = 0.02). For this analysis, length of stay was defined as time from admission to *C. auris* isolation, discharge or death, whichever came first. Non-invasive ventilation (BIPAP and HFNC) and mechanical ventilation were not reported as risk factors for isolating *C. auris* (*p* = 0.051 and *p* = 0.86 respectively). 

No statistically significant difference was observed among the two patient groups with regard to underlying risk factors such as diabetes, chronic kidney and lung diseases, active hematological and solid malignancies, chemotherapy, or active immunosuppressive therapy. In addition, pharmacological treatment (baricitinib, tocilizumab, and antibiotics) did not differ between the two groups. Both groups of severely ill COVID-19 patients with *C. auris* and COVID-19 patients without *C. auris* had similar percentages of superimposed MDR bacterial organisms (40.7% and 41.5%) (Table 2).

In the multiple logistic regression analysis, the variables that were previously shown in the literature to increase the risk of *C. auris* isolation (Candida score, DM II, non-invasive ventilation, mechanical ventilation, baricitinib, and tocilizumab) were included in our analysis in addition to qSOFA score and LOS. CVC and urinary catheter variables were excluded because a large proportion of patients in both Groups (*C. auris*+/COV+) and (*C. auris−*/COV+) had CVC (93.8% vs. 72.3% respectively) and urinary catheters (100% vs. 85.4% respectively). LOS (*p* = 0.008), and qSOFA score (*p* < 0.001) were the only independent risk factors associated with *C. auris* isolation (Table 3).

### 3.3. Mortality of Patients with C. auris According to the Site of Isolation

A total of 56 patients had *C. auris* isolated from bloodstream, DTA, urine, and wound, and 2 of them grew *C. auris* from blood and DTA concomitantly, making the total number of isolates equal to 58. Eight patients out of 56 had bloodstream infections with *C. auris* and 75% of them died during their hospital stay. They all had a urinary catheter and 7 of them had a central venous catheter. Their mean age was 64.3 years and the median of their length of stay was 57 days. Only 3 of them had COVID-19 infection. Twenty-six patients grew *C. auris* from DTA, 11 died (42.3%). Twenty patients grew *C. auris from* urine, 11 died (55%); 4 patients had *C. auris* isolated from wounds and 2 of them died (40%) (Table 4). 

### 3.4. Impact of C. auris on Death within 30 Days of Its Isolation (Early In-Hospital Mortality)

The impact of *C. auris* on death within 30 days of its isolation was analyzed and compared in all patients admitted to the ICU during that period regardless of their COVID-19 status, meaning (*C. auris*+/COV−) and (*C. auris*+*/*COV+), and is presented in Table 5. A mortality of 50% within 30 days of isolating *C. auris* was noted among both groups. For this analysis, length of stay was defined as the period of time between admission and discharge or death in days. The factors associated with increased mortality within 30 days of *C. auris* isolation were LOS of 72.6 days +/− 57.1 (vs. 43.7 +/− 19.3 in those who lived) (*p* = 0.02), a Candida score of 2.4 +/− 0.9 (vs. 1.54 +/− 1.3 among those who survived) (*p* = 0.01), and the occurrence of septic shock in 96.4% of patients who died (vs. 35.7% of patients who survived) (*p* < 0.001). Microbiological eradication did not have an impact on early in-hospital mortality (*p* = 0.51); 67.9% of the patients who survived till day 30, compared to only 39.3% of the patients who died by day 30, received antifungal therapy for more than 7 days after *C. auris* isolation (*p* = 0.03). Antifungal treatment was given to patients in whom there was suspicion of invasive fungal infection, and it was mostly empiric therapy as most of our isolates were saved for later susceptibility testing after the pandemic settled down amidst the ongoing economic crisis in Lebanon.

COVID-19-related treatments did not impact early in-hospital mortality; in the Group (*C. auris*+/COV+) where tocilizumab and barcitinib were given for management of cytokine storms (n = 32), there was no statistical significance in mortality, with 62.5% alive and 68.8% death for those taking tocilizumab, and 18.8% alive and 18.8% death for those taking baricitinib. No difference was found in mortality among patients taking prednisone, with 64.3% alive and 71.4% death within 30 days of *C. auris* isolation. Antibiotics were administered to all patients and therefore, no analysis was done. However, among the types of antibiotics, a higher mortality was noted in the patients who received tigecycline compared to other antibiotics (57.1% of patients who died at day 30 compared to 17.9% of patients who stayed alive at day 30, *p* = 0.01). The administration of other antibiotics, the isolation of MDR bacterial pathogens, the studied comorbidities, the presence of CVC and urinary catheter, non-invasive mechanical ventilation, and mechanical ventilation did not have any significant impact on early in-hospital mortality (Table 5).

In the multiple logistic regression analysis, the variables that have been shown in the literature to affect early in-hospital mortality (Candida score, DM II, non-invasive ventilation, mechanical ventilation, baricitinib, tocilizumab, qSOFA score, appropriate antifungal therapy, LOS, and qSOFA) were analyzed. No previous study from the literature was done about the association of the use of tigecycline and mortality from *C. auris*. In addition, a small number of our patients received tigecycline. Thus, this antibiotic was excluded from the multivariable analysis model. Septic shock was the only independent risk factor to increase mortality in patients who acquired *C. auris* (*p* = 0.006).

### 3.5. Molecular Characterization of C. auris Isolates

The reported antifungal susceptibilities on 28 *C. auris* isolates are shown in Table 6 [22]. *C. auris* was considered to be uniformly resistant to fluconazole, and 100% of the isolates demonstrated specified breakpoints of resistance (MIC ≥ 2 ug/mL) for amphotericin B. On the contrary, there was a uniform susceptibility of the isolates for micafungin and caspofungin [22].

As for the drug-resistance-associated genes, 2 mutations were detected: the lanosterol 14-α-demethylase-encoding gene *ERG11* (Y132F) and a novel mutation D709E, found in *CDR1* gene that encodes for an ABC efflux pump [22].

The genomes that were generated were compared to all *C. auris* genomes available in the National Center for Biotechnology (NCBI) database and the results indicated that the *C. auris* isolates belonged to South Asian Clade I.

## 4. Discussion

Several published reports showed worldwide hospital outbreaks of *C. auris* during the COVID-19 pandemic [8,11,24]. In our hospital, *C. auris* was first isolated in October 2020, and the horizontal spread continued since then, despite the rigorous IPC measures that were implemented. The possible etiologies of our hospital outbreak might have been linked to an initial case transferred to our facility by a private jet from Africa, and the spread due to the tremendous pressure on the staff in our medical center during the pandemic and the large number of patients in critical care units and ED, which resulted in suboptimal IPC practices. Many reports from the literature showed a similar horizontal transmission of *C. auris* during the COVID-19 pandemic [9,11,20]. The organism can be transmitted on the hands of healthcare workers or on shared equipment. A study from Salvador, Brazil, showed that axillary thermometers may have facilitated the dissemination of *C. auris* in the COVID-19 ICU [25]. Aggressive investigation was conducted to identify the source of *C. auris* at AUBMC. During our study period, a decrease in transmissions was noted following the implementation of new and rigorous IC measures that included placing patients with *C. auris* on contact isolation and practicing hand hygiene with soap and water followed by alcohol-based solutions after exiting the patient’s room. The cleaning and disinfecting of the patient care environment was changed to hypochlorite-based disinfectants, since brands containing quaternary ammonia compounds (QACs) are not effective against *C. auris* [5]. Sharing of medical supplies and equipment was prohibited when possible. As for the items that had to be shared between patients, thorough cleaning and disinfection practices were implemented between patients. Terminal decontamination of the rooms using vaporized hydrogen peroxide (H_2_O_2_) was conducted after the patient’s discharge. Despite all these efforts, transmissions could not be completely stopped, and this was most likely caused by understaffing of units and increase in the patients to nursing ratios, making adherence to rigorous IC practices at times difficult to achieve.

Our study is one of the first few studies tackling specifically the risk factors of isolating *C. auris* in severe COVID-19 patients, and its outcomes in severe COVID-19 and non-COVID-19 ICU patients. The bivariate regressions analysis showed that the presence of urinary catheter or CVC, high qSOFA score, and prolonged hospital stay were associated with *C. auris* isolation in severe COVID-19 patients. In the multivariable logistic regression analysis, only LOS and high qSOFA score were shown to be independent risk factors for *C. auris* isolation. Longer ICU stay had been shown to be a risk factor for the development of candidemia, including *C. auris* fungemia in several studies, possibly because of horizontal transmission of *C. auris* during a prolonged ICU stay [10,26,27,28]. In a case-control study conducted by Omrani et al. to assess risk factors for candidemia, including *C. auris* in COVID-19 patients, 8 out of 82 cases of candidemia were due to *C. auris*, and qSOFA score in addition to age were shown to be the only independent risk factors for candidemia in COVID-19 patients admitted to the ICU [29]. Another case-control study from 2 hospitals in India assessing risk factors of candidemia in severe COVID-19 patients showed that *C. auris* was the predominantly isolated species followed by *C. tropicalis*, and that prolonged ICU stay and raised ferritin level were independent predictors for the development of candidemia [30]. We did not look at the ferritin level in our study since not all our patients had ferritin measurement. A report from Mexico showed that all patients with isolated *C. auris* had CVC or urinary catheter, which is similar to what was found in our bivariate analysis [8]. Since source control is an integral part of the management of bloodstream and urinary candidiasis, including those caused by *C. auris*, it is highly recommended to remove the urinary catheter or CVC in *C. auris*-infected patients [30,31]. In our series, all patients with *C. auris* isolated from blood had their CVC removed.

Immunosuppressive agents such as steroids and tocilizumab are widely used in severe COVID-19 patients and have been shown to reduce mortality in those patients [32]. Previous studies showed that patients with COVID-19 who received tocilizumab were more likely to have superimposed fungal infections [33,34]. Although in our study, a higher percentage of patients with severe COVID-19 and *C. auris* had used tocilizumab compared to patients with severe COVID-19 alone (65.9% vs. 49.2%), tocilizumab was not found to be a significant risk factor for *C. auris* isolation. This result was similar to the study performed in Qatar where no association was found between candidemia, including *C. auris*, and the use of corticosteroids and tocilizumab after matching for the period of LOS in the ICU [29]. On the contrary, in a study conducted by Rajni et al., the multivariate logistic regression analysis showed that tocilizumab was an independent predictor of candidemia in severe COVID-19 patients [10].

In a study by Moein et al., patients with *C. auris* candidemia had higher rate of isolation of MDR bacteria, but this was not statistically significant [26]. In fact, bacterial superinfections with SARS-CoV-2 have been widely reported in the literature, and they usually occur in patients requiring ventilation and prolonged ICU stay [4,35,36,37]. However, whether superimposed bacterial infections will increase the risk of acquiring superimposed candida infections is to be explored in future studies. 

This is one of the first few studies in the literature to evaluate the impact of *C. auris* on mortality in patients with severe COVID-19. Our analysis showed that 30-day mortality was 50% in all patients with isolated *C. auris* in the setting of severe COVID-19 infection. The bivariate analysis showed that the risk factors associated with higher mortality were prolonged LOS, septic shock, higher Candida score, and antifungal therapy given for at least 7 days. However, the multivariable regression analysis showed that only septic shock was shown to be an independent risk factor between *C. auris* and mortality in patients with severe COVID-19. In an international multicenter study of *C. auris* by Pandya et al., where they included patients with *C. auris* isolated from blood, respiratory tract, skin and soft tissues, and urine, the overall mortality was 37% [30]. A higher Candida score was shown to be a risk factor for *C. auris* candidemia in the ICU of a tertiary care center in Spain [38], and in our study, it was also shown to affect mortality. Furthermore, in terms of management strategies, in our study, antifungal treatment for at least 7 days was associated with a significant reduction in 30-day mortality, which is similar to what was found in other studies [30]. Although not all patients in our series had their *C. auris* susceptibility testing done for financial reasons, since the first several isolates were susceptible to echinocandins and resistant to azoles, all the patients with isolated *C. auris* received echinocandins when treatment was deemed necessary.

It is worth mentioning that evidence of microbiological eradication in our study did not have any impact on improved mortality. In an era where antifungal stewardship is recommended, this finding further reinforces that antifungal therapy should only be used in patients with suspected invasive fungal infection and not to eradicate colonization with *C. auris*. In fact, it is believed that patients generally remain colonized with *C. auris* for a prolonged period of time and perhaps indefinitely [5]. Moreover, the CDC does not recommend treating *C. auris* isolated from noninvasive sites (urine, respiratory tract, and skin), if there is no suspicion for invasive fungal infection [5]. The application of such practice is challenging, since diagnosing invasive fungal infection in non-sterile sites in ICU patients (sites other than blood and CSF usually) is mostly guided by clinical judgment. In our study, patients had severe COVID-19 and prolonged ICU stay, thus, it was difficult to know whether their fever and clinical deterioration were attributed to the isolated *C. auris* causing invasive fungal infection or to the progression of COVID-19. Unfortunately, because of the economic crisis in Lebanon, 1,3-β-D-Glucan is not currently available nationally to help guide our management. 

It is interesting to note that only 8 patients had *C. auris* candidemia while 48 had non-candidemia infections. As expected, patients with candidemia had a higher mortality; however, this was not statistically significant. Nevertheless, it is worth mentioning that the number of candidemia patients was small and this could have affected our analysis. Sites of isolation were mostly DTA, followed by urine cultures. Most of the reported cases of *C. auris* in the literature were isolated from blood [30,39,40]. One study from Saudi Arabia showed that the most commonly isolated site of *C. auris* was urine (42.9%), followed by blood (17.1%) [41].

The susceptibilities of our *C. auris* isolates were similar to those reported from our region [42,43,44,45] and phylogenetically, the isolates’ genomes belonged to the South Asian clade I. This clade was also found in other countries of the same region: Oman, Qatar, and Saudi Arabia [46]. The phylogenetic analysis showed a limited genetic diversity which implied in-hospital transmission of this pathogen [22], in accordance with what was suggested earlier by a study published from AUBMC [11]. 

The strength of our study is the wide clinical data and clinical characteristics presented on patients with acquired *C. auris*. In addition, our study is the first case-control study in the literature evaluating specifically *C. auris* isolation in severe COVID-19 patients admitted to the ICUs. However, several limitations are present. First, this is a retrospective study, therefore, bias could have happened while collecting data. Second, we did not have a well-matched control group, as all patients who fit the inclusion definitions during the study period were included, and this could have selected for possible confounders. Third, positive *C. auris* cultures done as part of infection control screening were excluded and this might have affected our analysis. Finally, because of the retrospective nature of the study, the number of patients was not equally distributed between the three groups which might have limited our analysis and conclusions. Future prospective studies with similar population characteristics are needed to draw better conclusions.

## 5. Conclusions

*C. auris* is a pathogen with significant infection control implications which has serious consequences on patients’ outcomes including those infected with COVID-19. The presence of a CVC, a urinary catheter, high qSOFA score, and a prolonged LOS were associated with increased *C. auris* isolation rates in patients with severe COVID-19 infection admitted to the ICUs. Prolonged LOS and qSOFA score were the only independent risk factors for *C. auris* isolation, whereas septic shock was the only identified independent risk factor that increased mortality in patients with severe COVID-19 infections and *C. auris.* These findings broaden our understanding of *C. auris* infections in the setting of COVID-19, and call for implementing rigorous infection prevention and control measures and antimicrobial stewardship practices while managing COVID-19-infected patients to curtail the spread of this pathogen. Future prospective multicenter studies are needed to better understand the scope of this infection. 

## Figures and Tables

**Table 1 microorganisms-10-01011-t001:** Demographics and characteristics of patients admitted to the ICU.

	(*C. auris*+/COV−)	(*C. auris*+/COV+)	(*C. auris−*/COV+)
	*C. auris* (n = 24)	*C. auris* and COVID-19 (n = 32)	COVID-19 (n = 130)
Patients’ Characteristics			
Age—mean (sd)	70.5 (14.9)	68.4 (14.2)	69.5 (12.2)
Male—no. (%)	10 (41.7)	19 (59.4)	91 (70.0)
qSOFA score—mean (sd)	1.58 (0.78)	1.63 (0.79)	1.06 (0.84)
Charlson score—mean (sd)	5.38 (2.16)	5.00 (2.87)	4.54 (2.48)
Candida score—mean (sd)	2.42 (1.38)	1.59 (0.98)	1.39 (1.17)
Length of stay in the hospital in days—mean (sd)	48.3 (30.9)	65.5 (51.9)	27.0 (17.6)
Comorbidities			
Diabetes mellitus—no. (%)	8 (33.3)	13 (40.6)	53 (40.8)
Chronic kidney disease—no. (%)	2 (8.3)	6 (18.8)	26 (20.0)
Chronic lung disease—no. (%)	5 (20.8)	7 (21.9)	19 (14.6)
Active hematological malignancy—no. (%)	3 (12.5)	3 (9.4)	7 (5.4)
Active solid malignancy—no. (%)	8 (33.3)	4 (12.5)	16 (12.3)
Chemotherapy within 30 days of *C. auris* or SARS-CoV-2 infection—no. (%)	1 (4.2)	4 (12.5)	12 (9.2)
Active immunosuppressive therapy within 30 days of *C. auris* or SARS-CoV-2 infection—no. (%)	3 (12.5)	2 (6.3)	9 (6.9)
Previous surgery within 30 days of *C. auris* isolation—no. (%)	12 (50.0)	6 (18.8)	-
Presence of indwelling devices			
Central venous catheter—no. (%)	20 (83.3)	30 (93.8)	94 (72.3)
Urinary catheter—no. (%)	24 (100)	32 (100)	111 (85.4)
Mechanical ventilation—no. (%)	16 (66.7)	27 (84.4)	108 (83.1)
Non-invasive ventilation—no. (%)	4 (16.7)	24 (75.0)	115 (88.5)
Nasogastric tube—no. (%)	19 (79.2)	28 (87.5)	96 (73.8)
Hemodialysis—no. (%)	6 (25.0)	8 (25.0)	37 (28.5)
Parenteral nutrition—no. (%)	8 (33.3)	1 (3.1)	11 (8.5)
Drugs often used for COVID-19			
Baricitinib—no. (%)	0 (0.0)	6 (18.8)	16 (12.3)
Tocilizumab—no. (%)	0 (0.0)	21 (65.6)	64 (49.2)
Prednisone—no. (%)	6 (25.0)	32 (100.0)	130 (100.0)
Antibiotic use			
Any antibiotic use—no. (%)	24 (100)	32 (100.0)	128 (98.5)
Cephalosporin—no. (%)	14 (58.3)	20 (62.5)	73 (56.6)
Quinolone—no. (%)	4 (16.7)	19 (59.4)	91 (70.5)
Piperacillin-tazobactam—no. (%)	15 (62.5)	27 (84.4)	102 (79.1)
Carbapenem—no. (%)	18 (75.0)	23 (82.1)	20 (71.4)
Ceftazidime-avibactam—no. (%)	10 (41.7)	20 (62.5)	60 (46.5)
Aminoglycoside—no. (%)	7 (29.2)	22 (68.8)	98 (76.0)
Ceftolozane-tazobactam—no. (%)	2 (8.3)	7 (21.9)	25 (19.4)
Glycopeptide—no. (%)	12 (50.0)	24 (75.0)	100 (77.5)
Linezolid—no. (%)	1 (4.2)	9 (28.1)	20 (15.5)
Tigecycline—no. (%)	6 (25.0)	15 (46.9)	60 (46.5)
Colistin—no. (%)	8 (33.3)	8 (25.0)	33 (25.4)
Isolation of an MDR pathogen			
VRE—no. (%)	3 (12.5)	3 (9.4)	11 (8.5)
CRE—no. (%)	4 (16.7)	1 (3.1)	9 (6.9)
MRSA—no. (%)	0 (0.0)	2 (6.3)	5 (3.8)
MDR *Acinetobacter* spp.—no. (%)	0 (0.0)	0 (0.0)	7 (5.4)
MDR *Pseudomonas aeruginosa*—no. (%)	0 (0.0)	1 (3.1)	3 (2.3)
ESBL Enterobacterales—no. (%)	11 (45.8)	6 (18.8)	19 (14.6)
Outcome			
Discharged home—no. (%)	8 (33.3)	9 (28.1)	23 (17.7)
In hospital death—no. (%)	15 (62.5)	19 (59.4)	94 (72.3)

*C. auris: *Candida auris*;* (*C. auris*+/COV−), patients with *C. auris* isolates; (*C. auris*+/COV+), patients with severe COVID-19 and *C. auris* isolates; (*C. auris−*/COV+), patients with severe COVID-19. CRE: Carbapenem-resistant Enterobacterales; ESBL: Extended-spectrum-β-lactamase; ICU: Intensive unit care; MDR: Multi-drug resistant; MRSA: methicillin-resistant *Staphylococcus aureus*; qSOFA: quick sequential organ failure assessment; VRE: Vancomycin-resistant enterococci.

**Table 2 microorganisms-10-01011-t002:** Potential risk factors for isolating *C. auris* in patients with severe COVID-19 admitted to the ICU in bivariable analysis.

	(*C. auris*+/COV+)	(*C. auris−*/COV+)	*p*-Value
	(n = 32)	(n = 130)	
Patients’ Characteristics			
Age—mean (sd)	68.44 (14.21)	69.52 (12.17)	0.66
Male—no. (%)	19 (59.4)	91 (70.0)	0.25
qSOFA score—mean (sd)	1.63 (0.79)	1.06 (0.84)	<0.001 *
Charlson score—mean (sd)	5.00 (2.87)	4.54 (2.48)	0.36
Candida score—mean (sd)	1.59 (0.98)	1.39 (1.17)	0.35
Comorbidities and Risk Factors			
Diabetes mellitus—no. (%)	13 (40.6)	53 (40.8)	0.99
Chronic kidney disease—no. (%)	6 (18.8)	26 (20.0)	0.87
Chronic lung disease—no. (%)	7 (21.9)	19 (14.6)	0.32
Active hematological malignancy—no. (%)	3 (9.4)	7 (5.4)	0.42
Active solid malignancy—no. (%)	4 (12.5)	16 (12.3)	0.68
Chemotherapy within 30 days—no. (%)	4 (12.5)	12 (9.2)	0.58
Active immunosuppressive therapy—no. (%)	2 (6.3)	9 (6.9)	0.89
Previous surgery within 30 days—no. (%)	6 (18.8)	-	-
ICU stay			
Central venous catheter—no. (%)	30 (93.8)	94 (72.3)	0.01 *
Urinary catheter—no. (%)	32 (100)	111 (85.4)	0.015 *
Mechanic ventilation—no. (%)	27 (84.4)	108 (83.1)	0.86
Non-invasive ventilation—no. (%)	24 (75.0)	115 (88.5)	0.051
Nasogastric tube—no. (%)	28 (87.5)	96 (73.8)	0.10
Parenteral nutrition—no. (%)	1 (3.1)	11 (8.5)	0.30
Hemodialysis—no. (%)	8 (25.0)	37 (28.5)	0.70
Length of stay in the hospital—mean (sd)	35.3 (20.5)	27.0 (17.6)	0.02 *
Drugs often used for COVID-19			
Baricitinib—no. (%)	6 (18.8)	16 (12.3)	0.34
Tocilizumab—no. (%)	21 (65.6)	64 (49.2)	0.10
Prednisone—no. (%)	32 (100.0)	130 (100.0)	-
Antibiotic use			
Antibiotic use—no. (%)	32 (100.0)	128 (98.5)	0.48
Cephalosporin—no. (%)	20 (62.5)	73 (56.2)	0.56
Quinolone—no. (%)	19 (59.4)	91 (70.0)	0.22
Piperacillin-tazobactam—no. (%)	27 (84.4)	101 (77.7)	0.50
Carbapenem—no. (%)	25 (78.1)	105 (80.7)	0.68
Ceftazidime-avibactam—no. (%)	20 (62.5)	60 (46.2)	0.11
Aminoglycoside—no. (%)	22 (68.8)	98 (75.4)	0.40
Ceftolozane-tazobactam—no. (%)	7 (21.9)	25 (19.2)	0.75
Glycopeptide—no. (%)	24 (75.0)	100 (76.9)	0.76
Linezolid—no. (%)	9 (28.1)	19 (14.6)	0.18
Tigecycline—no. (%)	15 (46.9)	60 (46.2)	0.97
Colistin—no. (%)	8 (25.0)	33 (25.4)	0.96
MDR pathogen			
VRE—no. (%)	3 (9.4)	11 (8.5)	0.87
CRE—no. (%)	1 (3.1)	9 (6.9)	0.42
MRSA—no. (%)	2 (6.3)	5 (3.8)	0.55
MDR *Acinetobacter* spp.—no. (%)	0 (0.0)	7 (5.4)	0.18
MDR *Pseudomonas aeruginosa*—no. (%)	1 (3.1)	3 (2.3)	0.79
ESBL—no. (%)	6 (18.8)	19 (14.6)	0.56

*C. auris: *Candida auris**; Group (*C. auris+*/COV+), patients with severe COVID-19 and *C. auris* isolates; Group (*C. auris−*/Cov+), patients with severe COVID-19. CRE: Carbapenem-resistant Enterobac-terales; ESBL: Extended-spectrum-β-lactamase; ICU: Intensive unit care; MDR: Multi-drug resistant; MRSA: methicillin-resistant *Staphylococcus aureus*; qSOFA: quick sequential organ failure assess-ment; VRE: Vancomycin-resistant enterococci. *: *p*-value < 0.05.

**Table 3 microorganisms-10-01011-t003:** Potential risk factors for isolating *C. auris* in patients with severe COVID-19 admitted to the ICU in multivariable analysis.

	Unadjusted OR	*p*-Value	Adjusted OR	*p*-Value
qSOFA	0.12 (0.05–0.29)	<0.001	0.108 (0.04–0.29)	<0.001
Length of stay in the hospital	–	0.02	1.024 (1.00–1.05)	0.008

ICU: Intensive unit care; qSOFA: quick sequential organ failure assessment; OR: Odds ratio.

**Table 4 microorganisms-10-01011-t004:** Mortality of patients with *C. auris* according to the site of isolation.

	Sites of *C. auris* Isolation (Total 58) *			Mortality—no. (%)		
	(*C. auris*+/COV−)	(*C. auris*+/COV+)	Total	(*C. auris*+/COV−)	(*C. auris*+/COV−)	Total
Bloodstream	5	3	8 (13.8)	4 (80)	2 (66.7)	6 (75)
DTA	9	17	26 (44.8)	4 (44.4)	7 (41.2)	11 (42.3)
Urine	10	10	20 (34.7)	5 (50)	6 (60)	11 (55)
Wound	1	3	4 (6.8)	0 (0)	2 (66.7)	2 (50)

* Two patients had *C. auris* isolated from bloodstream and DTA; DTA: deep tracheal aspirates; *C. auris: *Candida auris*.*

**Table 5 microorganisms-10-01011-t005:** Impact of *C. auris* isolation on mortality in patients with and without severe COVID-19 admitted to the ICU.

	Alive (n = 28)	Death (n = 28)	*p*-Value
Patients’ Characteristics			
Male—no. (%)	15 (53.6)	14 (50.6)	0.79
Age—mean (sd)	66.14 (13.7)	72.54 (14.6)	0.10
qSOFA score—mean (sd)	1.46 (0.8)	1.75 (0.8)	0.18
Charlson score—mean (sd)	4.64 (2.7)	5.68 (2.4)	0.13
Candida score—mean (sd)	1.54 (1.3)	2.4 (0.9)	0.01 *
Comorbidities and Risk Factors			
Diabetes mellitus—no. (%)	9 (32.1)	12 (42.9)	0.41
Chronic kidney disease—no. (%)	2 (7.1)	6 (21.4)	0.13
Chronic lung disease—no. (%)	6 (21.4)	6 (21.4)	1.00
Active hematological malignancy—no. (%)	3 (10.7)	3 (10.7)	1.00
Active solid malignancy—no. (%)	6 (21.4)	6 (21.4)	1.00
Chemotherapy within 30 days—no. (%)	1 (3.6)	4 (14.3)	0.16
Active immunosuppressive therapy—no. (%)	1 (3.6)	4 (14.3)	0.16
Previous surgery within 30 days—no. (%)	8 (28.6)	10 (35.7)	0.57
Central venous catheter—no. (%)	26 (92.9)	24 (85.7)	0.39
Urinary catheter—no. (%)	28 (100.0)	28 (100.0)	-
Mechanic ventilation—no. (%)	21 (75.0)	22 (78.6)	0.75
BIPAP—no. (%)	11 (39.3)	15 (53.6)	0.28
High flow nasal cannula—no. (%)	8 (28.6)	3 (10.7)	0.09
Non-invasive ventilation—no. (%)	13 (46.4)	15 (53.6)	0.59
Nasogastric tube—no. (%)	21 (75.0)	26 (92.9)	0.07
Parenteral nutrition—no. (%)	5 (17.9)	4 (14.3)	0.72
Hemodialysis—no. (%)	5 (17.9)	9 (32.1)	0.22
COVID-19 infection—no. (%)	16 (57.1)	16 (57.1)	1.00
Treatment			
Antifungal therapy for more than 7 days after *C. auris* isolation—no. (%)	19 (67.9)	11 (39.3)	0.03 *
Prednisone—no. (%)	18 (64.3)	20 (71.4)	0.57
Antibiotic use			
Antibiotic use—no. (%)	28 (100.0)	28 (100.0)	-
Cephalosporin—no. (%)	17 (60.7)	17 (60.7)	1.00
Quinolone—no. (%)	13 (46.4)	10 (35.7)	0.42
Piperacillin-tazobactam—no. (%)	21 (75.0)	21 (75.0)	1.00
Carbapenem—no. (%)	23 (82.1)	20 (71.4)	0.34
Ceftazidime-avibactam—no. (%)	14 (50.0)	16 (57.1)	0.59
Aminoglycoside—no. (%)	15 (53.6)	14 (50.0)	0.79
Ceftolozane-tazobactam—no. (%)	7 (25.0)	2 (7.1)	0.07
Glycopeptide—no. (%)	18 (64.3)	18 (64.3)	1.00
Linezolid—no. (%)	5 (17.9)	5 (17.9)	1.00
Tigecycline—no. (%)	5 (17.9)	16 (57.1)	0.01 *
Colistin—no. (%)	10 (35.7)	6 (21.4)	0.24
Isolation of MDR pathogen			
VRE—no. (%)	3 (10.7)	3 (10.7)	1.00
CRE—no. (%)	3 (10.7)	2 (7.1)	0.64
MRSA—no. (%)	0 (0.0)	2 (7.1)	0.15
MDR Acinetobacter spp.—no. (%)	0 (0.0)	0 (0.0)	–
MDR *Pseudomonas aeruginosa*—no. (%)	0 (0.0)	1 (3.6)	0.31
ESBL Enterobacterales—no. (%)	8 (28.6)	9 (32.1)	0.77
Patients’ outcome			
Septic shock—no. (%)	10 (35.7)	27 (96.4)	<0.001 *
Microbiologic eradication—no. (%)	7 (26.9)	5 (19.2)	0.74
Length of stay in the hospital in days—mean (sd)	72.6 (57.1)	43.7 (19.3)	0.02 *

*C. auris: *Candida auris*;* CRE: Carbapenem-resistant Enterobacterales; BIPAP: Bilevel positive airway pressure; ESBL: Extended-spectrum-β-lactamase; MDR: Multi-drug resistant; MRSA: methicillin-resistant *Staphylococcus aureus*; qSOFA: quick sequential organ failure assessment; VRE: Vancomycin-resistant enterococci. *: *p*-value < 0.05.

**Table 6 microorganisms-10-01011-t006:** Reported antifungal susceptibility results of 28 *C. auris* isolates from AUBMC.

Antifungal Agent	Susceptibility Findings					
	MIC50	MIC90	MIC Range	%S	%I	%R
Fluconazole	32	≥32	16–≥32	0	0	0
Voriconazole	0.25	0.25	0.12 to 4	36	61	3
Caspofungin	0.25	0.25	0.25–0.25	100	0	0
Micafungin	0.12	0.12	0.064–0.12	100	0	0
Amphotericin B	8	8	2–16	0	0	0

S: Susceptible; I: Intermediate; R: Resistant; AUBMC: American University of Beirut Medical Center; MIC: Minimal inhibitory concentration.

## Data Availability

The data presented in this study are available on request from the corresponding author. The data are not publicly available as per our IRB regulations.

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
