# Peer review of "COVID-19 and C. auris: A Case-Control Study from a Tertiary Care Center in Lebanon"

_microorganisms, 2022, doi:10.3390/microorganisms10051011_

Round 1

Reviewer 1 Report

The article is well thought out. Three groups of patients, one with C.auris without covid, another with C.auris,
another with C.auris and COVID, and another with COVID. The numbers of patients included in each group are differents and not very comparable,
as a result of retrospective studies. This is one of the fundamental problems of the study, it allows fewer conclusions to be drawn than would
be possible in a prospective study with groups similar in number
and conditions of the patients included.
The variables studied are correct and so is the statistical study.
The inclusion of tigecycline to which they give statistical value could be explained a little more.

Reviewer 2 Report

Overall comments (but more details in the document attached)

  • This seems to be more of a cohort study, not a case-control since there is not matching of cases and controls
  • Why were colonized patients identified through skin swabs excluded?
  • Some language about time frame of variables is not clear
  • The definition of microbiological eradication is clearly stated, but literature suggests that 2 negative cultures is likely not sufficient to determine eradication. https://pubmed.ncbi.nlm.nih.gov/32291441/
  • I would be careful using the word infection (Ln137 but throughout) when there may not have been sign/symptoms of infection as stated in the methods
  • Description of study population using group A, B, C was tough to understand the results and names that better reflected the population would make this easier
  • some terms throughout were not clear
  • the organization and placement of the tables should be improved to describe and compare groups
  • mortality analyses not clearly described in results, suggests evaluating risk factors between group A and group B, but groups were combined and comparison was between alive and dead
  • results from candidemia references (all species) is not directly comparable
  • should describe what is considered 'adequate antifungal therapy'

Reviewer 3 Report

General comments

The Authors performed a retrospective study in patients with severe COVID-19 infection with and without C. auris infection/colonization. They found that prolonged LOS and qSOFA score were independent risk factors for C. auris isolation. Moreover, septic shock was independent risk factor that increased mortality in patients with severe COVID-19 infections and C. auris.

Although the MS is interesting, data were mainly confirmatory.

Minor comments

  1. Lines 148-149. Is chocholate agar a routine medium for fungal subculture?
  2. Lines 222-228. Data may be better in a new Table.
  3. Line 343 and lines 351-352. I did not see where Authors have found inappropriate antifungal therapy.
  4. Lines 353-355. It would be worth to show the available MIC data in new Table in the “Results” section.
  5. Lines 380-384. These data should be presented in the “Results” section.
